# Differences in birth weight between immigrants' and natives' children in Europe and Australia: a LifeCycle comparative observational cohort study

Sandra Florian ![ORCID],[1] Mathieu Ichou,[1] Lidia Panico,[1,2] Stéphanie Pinel-Jacquemin,[3] Tanja G M Vrijkotte,[4] Margreet W Harskamp-van Ginkel ![ORCID],[4] Rae-Chi Huang,[5] Jennie Carson ![ORCID],[6] Loreto Santa Marina Rodriguez,[7,8] Mikel Subiza-Pérez,[8,9] Martine Vrijheid,[9,10] Sílvia Fernández-Barrés,[10,11] Tiffany C Yang ![ORCID],[12] John Wright ![ORCID],[12] Eva Corpeleijn,[13] Marloes Cardol,[13] Elena Isaevska ![ORCID],[14] Chiara Moccia,[15] Marjolein N Kooijman,[16,17] Ellis Voerman,[16,17] Vincent Jaddoe,[16,17] Marieke Welten,[16,17] Elena Spada,[18] Marisa Rebagliato,[19,20] Andrea Beneito,[21] Luca Ronfani,[22] Marie-Aline Charles[23,24]

For numbered affiliations see end of article.

**Correspondence to**
Dr Sandra Florian;
sandra.florian@ined.fr

## ABSTRACT

**Objective** Research on adults has identified an immigrant health advantage, known as the 'immigrant health paradox', by which migrants exhibit better health outcomes than natives. Is this health advantage transferred from parents to children in the form of higher birth weight relative to children of natives?

**Setting** Western Europe and Australia.

**Participants** We use data from nine birth cohorts participating in the LifeCycle Project, including five studies with large samples of immigrants' children: Etude Longitudinale Française depuis l'Enfance—France (N=12 494), the Raine Study—Australia (N=2283), Born in Bradford—UK (N=4132), Amsterdam Born Children and their Development study—Netherlands (N=4030) and the Generation R study—Netherlands (N=4877). We include male and female babies born to immigrant and native parents.

**Primary and secondary outcome measures** The primary outcome is birth weight measured in grams. Different specifications were tested: birth weight as a continuous variable including all births (DV1), the same variable but excluding babies born with over 4500 g (DV2), low birth weight as a 0–1 binary variable (1=birth weight below 2500 g) (DV3). Results using these three measures were similar, only results using DV1 are presented. Parental migration status is measured in four categories: both parents natives, both born abroad, only mother born abroad and only father born abroad.

**Results** Two patterns in children's birth weight by parental migration status emerged: higher birth weight among children of immigrants in France (+12 g, p<0.10) and Australia (+40 g, p<0.10) and lower birth weight among children of immigrants in the UK (−82 g, p<0.05) and the Netherlands (−80 g and −73 g, p<0.001) compared with natives' children. Smoking during pregnancy emerged as a mechanism explaining some of the birth weight gaps between children of immigrants and natives.

## STRENGTHS AND LIMITATIONS OF THIS STUDY

⇒ This study includes data from nine cohorts in six countries, allowing for a comparison of varied national contexts and diverse immigrant groups.
⇒ Results are based on a large combined sample of more than 38 000 children.
⇒ The use of harmonised health outcomes and sociodemographic variables, as well as standardised statistical analyses enables a rigorous multinational comparative study.
⇒ Yet, some of the cohort samples are representative at the national level while others are at the local level, which limits the generalisability of the findings for certain cohorts.
⇒ Some of the most vulnerable, recently arrived immigrant groups may be underrepresented in certain cohorts, potentially biasing the results for these immigrant communities.

**Conclusion** The immigrant health advantage is not universally transferred to children in the form of higher birth weight in all host countries. Further research should investigate whether this cross-national variation is due to differences in immigrant communities, social and healthcare contexts across host countries.

## INTRODUCTION

Birth weight is an important health indicator that has been associated with lifelong development, morbidity and mortality.[1–4] Studies have found a U-shape relationship between birth weight and the risk of developing health problems, including cardiometabolic complications, with both low and high birth weight associated with a higher risk.[4]

Research on adults has identified an immigrant health advantage, known as the 'immigrant health paradox',[5–7] by which migrants appear to have better health outcomes, including lower morbidity and mortality as well as better mental health and fewer health risk behaviours, than their native counterparts, despite being on average more socioeconomically disadvantaged.[8] This health advantage has been hypothesised to be linked to positive premigration selectivity.[7] Studies using American data suggest that this health advantage can be transferred from parents to children, at least during the early years, finding healthier birth weight (ie, ≥2500 g and ≤4500 g) and lower probability of low birth weight among children of Hispanic immigrants in the USA[4 9 10] compared with native peers. This advantage does not appear to apply to second-generation parents and their children.[11]

There is a less research on these relationships in Europe, although the available evidence suggests similar patterns, that is, better birth health of children born to first-generation migrant mothers.[12] Little research has however investigated patterns in children's birth weight by parental migration status through a cross-country comparative lens, particularly in Europe.[13] Investigating inequality in children's birth weight according to parents' migration status is important given its potential to inform social and health policies. Differences in national origin and host country contexts (social policies, health systems, economic opportunities, etc) are likely to shape inequalities in children's birth outcomes as well as immigrant parents' abilities to invest in their children's health and transfer them health 'capital'.[2] Thus, we expect that children's birth weight by parental migrant status will vary across host countries.

Parental socioeconomic status is also an important determinant of children's health outcomes,[8 14 15] and, hence, a potential confounder of the relationship between migration status and birth weight. Children of less educated parents and with low family incomes are more likely to exhibit lower birth weight.[6 16 17] This has led researchers to highlight the importance of considering parental socioeconomic background when investigating differences in birth outcomes according to migrant status.[18] Although immigrants in developed countries tend to be socioeconomically disadvantaged on average, the level of this disadvantage differs across host countries; it is therefore even more critical to consider socioeconomic status in comparative research.[18]

Parental health behaviours also constitute an important possible mechanism that may contribute to health inequality at birth.[19] The deleterious effects of mother' smoking during pregnancy on children's health at birth are well established in the literature.[20–23] Research indicates that immigrant mothers appear to have fewer health risk behaviours, such as less smoking and lower alcohol consumption.[22] We explore the role of maternal health behaviour (smoking during pregnancy) as a potential mechanism shaping inequalities in birth weight across migrant groups.

This study focuses on the differences in birth weight between children of natives and children of immigrants using recent data from eight European countries and Australia to investigate three questions. First, do children of immigrants exhibit higher or lower birth weight relative to children of natives? Second, how do these associations vary across host countries? Third, can maternal smoking during pregnancy explain any of these differences?

## METHODS
### Data
Data come from the Horizon 2020-funded LifeCycle Project-EU Child Cohort Network,[24] which comprises 19 pregnancy and childhood cohorts in Europe and Australia investigating early life health and its determinants, starting from pregnancy. LifeCycle cohorts collected data on demographic and socioeconomic characteristics, including maternal health, marital status, living arrangements, parental education, family income and parental health behaviours, among other variables.[24]

A total of nine cohorts collected data on migration status and were able to harmonise variables on socioeconomic and migrant status associated with health at birth. The results presented in the main text are based on five cohorts with a sufficiently large sample size (>1700 children of immigrants) to conduct multivariate analyses. These cohorts include: the Etude Longitudinale Française depuis l'Enfance—ELFE (France), a nationally representative cohort study following 18 329 births in France to mothers aged 18 and over in 2011;[25] the Raine Study (Australia), a prospective study of 2788 babies collecting data starting in 1989 from pregnancy onwards;[26] the Born in Bradford—BiB study (UK) that includes data from 13 524 children born during 2007–2011 in Bradford;[27] the Amsterdam Born Children and their Development—ABCD study (Netherlands) including data from 11 474 pregnancies in Amsterdam between 2003 and 2004;[28] and the Generation R study (Netherlands) which enrolled 9153 mothers with a delivery date from April 2002 until January 2006, with a total of 9747 live born children.[29]

The other participating cohorts with smaller sample sizes (<1700 children of immigrants) included the NINFEA study (Italy),[30] the Piccolipiù study (Italy),[31] the INfancia y Medio Ambiente (Environment and Childhood) INMA study (Spain)[32] and the GECKO Drenthe study (Netherlands).[33] Because of smaller sample sizes, it is more difficult to draw conclusions from these cohorts, nevertheless, online supplemental appendix 1 presents the sample distribution and children's birth weight by parental migration status and maternal region of birth for participating cohorts with a small number of children of immigrants. Model results for these smaller-sample cohorts are presented in online supplemental appendix 2.

## Patient and public involvement

Participants were recruited prior to and during pregnancy, as well as in childhood. All data have been deidentified. Ethical and legal responsibility for data management and security is maintained by the source studies or home institutions.[24]

## Measures

The dependent variable is the child's birth weight, a continuous variable measured in grams. Different specifications of our dependent variable are tested (see the Data analysis section). Children's migration status is measured in four categories: second-generation children, with both parents born abroad; 2.5 generation children with a mother born abroad, and native-born father; 2.5 generation children with a father born abroad and a native-born mother; and natives (reference), which included children with both parents born in the host country. Mothers' region of birth was measured in 10 categories: host country (reference for each cohort study), Western Europe, Eastern Europe, Other Europe and Central Asia, East Asia and Pacific, South Asia, Middle East and North Africa, sub-Saharan Africa, Latin America and the Caribbean, and North America.

Child characteristics at birth that were used as control variables included: the child's sex; a binary indicator for multiple births (single birth, multiple birth); birth order measured with continuous variables indicating the mother's parity; and the child's gestational age (in days).

Basic controls for mother's characteristics included maternal height (cm) and pre-pregnancy weight (kg), measured as continuous variables.

We consider two indicators of socioeconomic background: mother's education and household income at the child's birth. Mother's level of education was measured in three categories: high (reference), medium and low education. Household income quintile was measured using a cohort-specific log of the equivalised total disposable household monthly income in 2011, predicted using pan-European Union Statistics on Income and Living Conditions data.[34] The Australian Raine Study and the Dutch ABCD cohort used a household income quartile measure (the fourth quartile being the richest), collected at ages 1 and 5, respectively.

Finally, we included a binary indicator of mother's smoking behaviour during pregnancy (0=no smoking during pregnancy, 1=any smoking during the pregnancy). Each of the participating cohorts conducted the variable harmonisation for all the variables, following strict step-by-step protocols (see https://pubmed.ncbi.nlm.nih.gov/33884544/). Quality control checks were conducted to ensure the correct harmonisation of variables across cohorts.

## Data analysis

After descriptive statistics analyses, we selected our analytical sample following a complete-case analysis approach. We used ordinary least square and logistic regressions to model different specifications of birth weight. First, we modelled birth weight as a continuous variable, including all births (DV1). Second, we excluded babies born with a birth weight higher than 4500 g (DV2) given their higher risk of negative health outcomes. Finally, we modelled low birth weight as a 0–1 binary variable where 1 indicates birth weight below 2500 g (DV3). As the results for these three models were very similar, only DV1 is presented in the Results section. Models using the other specifications are available as online supplemental appendices 3 and 4.

Analyses were conducted separately for each cohort. Each cohort conducted two sets of models for all three birth weight specifications. The first set predicted birth weight regressed on children's migration status, while the second set regressed birth weight on maternal region of birth. Each set of models included three nested models. Model 1 (M1) controlled for basic characteristics at birth, including the child's sex, birth order, gestational age, and the mother's height and pre-pregnancy weight. Model 2 (M2) additionally controlled for family socioeconomic characteristics, including the mothers' education and family income. Model 3 (M3) further adjusted for mothers' smoking during pregnancy.

## RESULTS
### Birth weight differences by parental migration status

Table 1 presents the sample distribution and the average birth weight for children by parental migration status and maternal region of birth for the five largest cohorts, as well as the distribution of cases excluded from the analysis due to missing values. As table 1 shows, immigrant group representation varies by cohort. The largest immigrant groups in the Raine Study (AU) come from Western European and East Asian and Pacific countries. For ELFE (FR) the largest groups are immigrants from Middle East and North Africa followed by sub-Saharan African. For BiB (UK) the largest group come from South Asia; whereas in ABCD (NL) and Generation R (NL) the groups with the largest representations come from Latin America, the Caribbean as well as Middle East and North Africa.

Table 2 presents the first set of models predicting birth weight regressed on children's migration status and figures 1 and 2 illustrate the results of table 2. We observed two distinct patterns regarding the disparities of children's weight at birth by parental migration status in the five largest cohorts (figures 1 and 2, table 2). The 'small sample' cohorts exhibited similar patterns, but the differences were not statistically significant, possibly due to small samples of children of immigrants (online supplemental appendix 2).

In the first pattern, depicted by two cohorts, ELFE (France) and Raine (Australia), children of immigrants exhibited on average higher weight at birth relative to children of natives, as shown in figure 1. For ELFE, M1 (grey bars) shows a slight tendency for children of immigrants to be born with higher weight than children with

**Table 1** Cohort sample distribution and children's birth weight (in grams) by migration status and mother's region of origin (large cohorts)

| | Raine Study (AU) | | | ELFE (FR) | | | BiB (UK) | | | ABCD (NL) | | | Gen R (NL) | | |
|---|---|---|---|---|---|---|---|---|---|---|---|---|---|---|---|
| | Freq. | % | Weight Mean | Freq. | % | Weight Mean | Freq. | % | Weight Mean | Freq. | % | Weight Mean | Freq. | % | Weight Mean |
| **Children's migration status** | | | | | | | | | | | | | | | |
| 2nd generation | 697 | 25.0 | 3314 | 1289 | 7.0 | 3349 | 2488 | 18.5 | 3186 | 2165 | 18.9 | 3313 | 1887 | 19.4 | 3321 |
| 2.5 gen. (immigrant mother) | 407 | 14.6 | 3303 | 918 | 5.0 | 3344 | 1595 | 11.7 | 3112 | 743 | 6.5 | 3393 | 767 | 7.9 | 3397 |
| 2.5 gen. (immigrant father) | 437 | 15.7 | 3363 | 1159 | 6.3 | 3348 | 1831 | 13.5 | 3120 | 943 | 8.2 | 3362 | 1040 | 10.7 | 3318 |
| Natives | 1137 | 40.8 | 3288 | 13 257 | 72.3 | 3308 | 5224 | 38.7 | 3293 | 3957 | 34.5 | 3455 | 4588 | 47.1 | 3452 |
| Missing | 110 | 3.9 | 2677 | 1706 | 9.3 | 3243 | 2386 | 17.6 | 3162 | 3666 | 32.0 | 3368 | 1465 | 15.0 | 3309 |
| **Total** | **2788** | **100** | **3284** | **18 329** | **100** | **3309** | **13 524** | **100** | **3205** | **11 474** | **100** | **3389** | **9747** | **100** | **3387** |
| **Mother's region of birth** | | | | | | | | | | | | | | | |
| Host country-born | 1608 | 57.7 | 3305 | 14 839 | 81.0 | 3307 | 7084 | 52.4 | 3247 | 4985 | 43.4 | 3435 | 5929 | 60.8 | 3419 |
| Western EU/EEA | 620 | 22.2 | 3344 | 278 | 1.5 | 3313 | 91 | 0.7 | 3256 | 401 | 3.5 | 3423 | 436 | 4.5 | 3408 |
| Eastern EU | 33 | 1.2 | 3414 | 101 | 0.6 | 3410 | 243 | 1.8 | 3388 | 67 | 0.6 | 3390 | 56 | 0.6 | 3333 |
| Other Europe and Central Asia | 0 | 0 | 0 | 140 | 0.8 | 3341 | 27 | 0.2 | 3490 | 390 | 3.4 | 3384 | 587 | 6.0 | 3422 |
| East Asia and Pacific | 300 | 10.8 | 3264 | 129 | 0.7 | 3354 | 121 | 0.9 | 3272 | 151 | 1.3 | 3409 | 200 | 2.1 | 3304 |
| South Asia | 55 | 2.0 | 3152 | 23 | 0.1 | 3162 | 3291 | 24.2 | 3118 | 145 | 1.3 | 3199 | 31 | 0.3 | 3393 |
| Middle East and North Africa | 15 | 0.5 | 3298 | 858 | 4.7 | 3392 | 78 | 0.6 | 3423 | 729 | 6.4 | 3444 | 533 | 5.5 | 3531 |
| Sub-Saharan Africa | 60 | 2.2 | 3219 | 604 | 3.3 | 3297 | 219 | 1.7 | 3232 | 344 | 3.0 | 3226 | 363 | 3.7 | 3236 |
| Latin America and Caribbean | 16 | 0.6 | 3320 | 125 | 0.7 | 3230 | 12 | 0.1 | 3011 | 666 | 5.8 | 3201 | 884 | 9.1 | 3200 |
| North America | 12 | 0.4 | 3404 | 18 | 0.1 | 3131 | 15 | 0.1 | 3354 | 67 | 0.6 | 3421 | 39 | 0.4 | 3391 |
| Missing | 69 | 2.5 | 2412 | 1214 | 6.6 | 3282 | 2343 | 17.3 | 3164 | 3529 | 30.8 | 3366 | 689 | 7.1 | 3300 |
| **Total** | **2788** | **100** | **3284** | **18 329** | **100** | **3309** | **13 524** | **100** | **3205** | **11 474** | **100** | **3389** | **9747** | **100** | **3387** |
| **Missing cases in analyses (n)** | n | | Mean | n | | Mean | n | | Mean | n | | Mean | n | | Mean |
| Birth weight | 80 | | – | 503 | | – | 334 | | – | 921 | | – | 90 | | – |
| Child and mother's controls | 112 | | 2520 | 531 | | 3301 | 8097 | | 3197 | 3307 | | 3370 | 2376 | | 3349 |
| Migration status | 44 | | 3178 | 1495 | | 3239 | 2386 | | 3162 | 148 | | 3304 | 636 | | 3326 |
| Missing other variables | 269 | | 3286 | 3306 | | 3261 | 3564 | | 3183 | 3068 | | 3311 | 1768 | | 3341 |
| **Total excluded cases** | **505** | | **2985** | **5835** | | **3259** | **9392** | | **3198** | **7444** | | **3344** | **4870** | | **3343** |

ABCD, Amsterdam Born Children and their Development; AU, Australia; BiB, Born in Bradford; EEA, European Economic Area; ELFE, Etude Longitudinale Française depuis l'Enfance; EU, European Union; Gen R, Generation R.

**Table 2** Ordinary least square regression coefficients of children's migration status on child's birth weight (large cohorts)

| DV1: birth weight | ELFE (FR) | | | Raine Study (AU) | | | BiB (UK) | | |
|---|---|---|---|---|---|---|---|---|---|
| | M1 | M2 | M3 | M1 | M2 | M3 | M1 | M2 | M3 |
| | Child controls | SES | Smoked | Child controls | SES | Smoked | Child controls | SES | Smoked |
| Children's migration status | | | | | | | | | |
| Natives (ref.) | | | | | | | | | |
| 2nd generation | 11.9 | 42.5* | 13 | 39.8 | 34.4 | 16.7 | −81.5*** | −83.4*** | −118.8*** |
| 2.5 generation (mother) | 47.6** | 53.9*** | 43.2** | 41.4 | 38 | 32.6 | −97.8*** | −94.7*** | −132.8*** |
| 2.5 generation (father) | 3.2 | 16.7 | 11.6 | 48.1 | 46.7 | 38.2 | −117*** | −120.8*** | −147.8*** |
| Child controls | | | | | | | | | |
| Female | −149.3*** | −149.2*** | −150.2*** | −130.2*** | −129.8*** | −123.3*** | −145.4*** | −143.5*** | −144.6*** |
| Plural birth† | −372.1*** | −374.2*** | −378*** | 0 | 0 | 0 | −311.2*** | −305*** | −309.9*** |
| Mother's parity (birth order) | 49.9*** | 54.9*** | 54*** | 64.8*** | 65.3*** | 64.7*** | 54.8*** | 62.7*** | 59.8*** |
| Gestational age | 23.7*** | 23.6*** | 23.5*** | 20*** | 20*** | 20.1*** | 25.3*** | 25.3*** | 25.2*** |
| Mother controls | | | | | | | | | |
| Height | 9*** | 8.5*** | 8.7*** | 11*** | 10.7*** | 11*** | 11.3*** | 10.9*** | 11.1*** |
| Pre-pregnancy weight | 4.7*** | 5*** | 4.9*** | 5.6*** | 5.7*** | 5.5*** | 5.6*** | 5.6*** | 5.6*** |
| SES | | | | | | | | | |
| Education‡ | | | | | | | | | |
| High (ref.) | | | | | | | | | |
| Medium | | −11 | −2.7 | | 16.4 | 18.5 | | −0.7 | 1.7 |
| Low | | −34.8* | −15.9 | | −1 | 19.5 | | −24.9 | −21.9 |
| Household income quintiles§ | | | | | | | | | |
| 1st quintile (ref.) | | | | | | | | | |
| 2nd quintile | | 52.3*** | 44.9*** | | −6.2 | 4.9 | | 1.7 | −9.7 |
| 3rd quintile | | 50.2*** | 32.9* | | −31.4 | −13.7 | | −38 | −22.9 |
| 4th quintile | | 57.3*** | 36.9* | | −30.6 | 2.3 | | −38.8 | −27.7 |
| 5th quintile | | 60.8*** | 36.4* | | – | – | | −68.6** | −31 |
| Mother smoked (pregnancy) | | | −118.6 | | | −151.6 | | | −135.7 |
| Constant | −5049.5*** | −4996.3*** | −4972.1*** | −4281.4*** | −4236.2*** | −4261.9*** | −5889.6*** | −5810.7*** | −5766.4*** |
| N | 12 494 | 12 494 | 12 494 | 2283 | 2283 | 2283 | 4132 | 4132 | 4132 |
| $R^2$ | 0.358 | 0.361 | 0.369 | 0.379 | 0.38 | 0.393 | 0.489 | 0.492 | 0.498 |

Continued

**Table 2** Continued

| DV1: birth weight | ABCD (NL) | | | Gen R (NL) | | |
|---|---|---|---|---|---|---|
| | M1 | M2 | M3 | M1 | M2 | M3 |
| | Child controls | SES | Smoked | Child controls | SES | Smoked |
| Children's migration status | | | | | | |
| Natives (ref.) | | | | | | |
| 2nd generation | −80.1*** | −52.6* | −66.5** | −73.4*** | −33.6 | −50.2* |
| 2.5 generation (mother) | 14.6 | 23.3 | 14.9 | 26.7 | 42.5 | 36.4 |
| 2.5 generation (father) | −78.3*** | −65.6** | −61.5** | −75.3*** | −53.9* | −51.1** |
| Child controls | | | | | | |
| Female | −121.9*** | −123.4*** | −123.6*** | −106.8*** | −107.2*** | −108.2*** |
| Plural birth† | 0 | 0 | 0 | −404.8*** | −408.5*** | −409.1*** |
| Mother's parity (birth order) | 82*** | 84.3*** | 83.1*** | 109.3*** | 111.8*** | 109.3*** |
| Gestational age | 25.7*** | 25.6*** | 25.6*** | 26.1*** | 26*** | 25.9*** |
| Mother controls | | | | | | |
| Height | 10.4*** | 9.5*** | 9.5*** | 10.1*** | 9.2*** | 9.3*** |
| Pre-pregnancy weight | 4.6*** | 5*** | 4.9*** | 4.5*** | 4.8*** | 4.7*** |
| SES | | | | | | |
| Education‡ | | | | | | |
| High (ref.) | | | | | | |
| Medium | | −54.9*** | −44.1** | | −10 | −7.3 |
| Low | | −59.7** | −38.2 | | −30.7 | −23 |
| Household income quintiles§ | | | | | | |
| 1st quintile (ref.) | | | | | | |
| 2nd quintile | | 2.7 | 5.9 | | 21.4 | 15.2 |
| 3rd quintile | | −3.9 | 5.9 | | 56.1* | 42.9 |
| 4th quintile | | −30.6 | −18.1 | | 77.1* | 56.4 |
| 5th quintile | | – | – | | 84.2* | 63.1 |
| Mother smoked (pregnancy) | | | −135.5*** | | | −79.5*** |
| Constant | −5733.5*** | −5573.2*** | −5550.5*** | −5951.2*** | −5823.6*** | −5780.1*** |
| N | 4030 | 4030 | 4030 | 4877 | 4877 | 4877 |

Continued

**Table 2** Continued

| | ABCD (NL) | | | Gen R (NL) | | |
|---|---|---|---|---|---|---|
| | M1 | M2 | M3 | M1 | M2 | M3 |
| **DV1: birth weight** | Child controls | SES | Smoked | Child controls | SES | Smoked |
| $R^2$ | 0.401 | 0.404 | 0.409 | 0.46 | 0.46 | 0.46 |

M1: basic child and mother's controls at birth. M2: M1 + socioeconomic status (SES) variables. M3: M2 + mother's smoking during pregnancy.

*p<0.05, **p<0.01, ***p<0.001.

†The ABCD and the Raine Study cohorts only included single births in their sample.

‡Level of education based on the highest ongoing or completed education when the child was 0 years old (between >−1 year and <1 year). If more than one education level is reported within the defined time frame, we used the highest recorded education level. Classification according to International Standard Classification of Education 97/2011 (ISCED-97/2011). High: short cycle tertiary, bachelor, masters, doctoral or equivalent (ISCED-2011: 5-8, ISCED-97: 5-6). Medium: upper secondary, post-secondary non-tertiary (ISCED-2011: 3-4, ISCED-97: 3-4). Low: no education; early childhood; pre-primary; primary; lower secondary or second stage of basic education (ISCED-2011: 0-2, ISCED-97: 0-2).

§European Union Statistics on Income and Living Conditions income quintiles for all cohorts except for Raine (AU) and ABCD (NL) that measured household income in quartiles.

ABCD, Amsterdam Born Children and their Development; AU, Australia; BiB, Born in Bradford; ELFE, Etude Longitudinale Française depuis l'Enfance; Gen R, Generation R.

both native parents only among 2.5 generation, immigrant mothers (+48 g, p<0.001). For the ELFE cohort, adjusting for socioeconomic characteristics in M2 (white bars), children of immigrants' weight *increased* compared with natives and was significant for the 2nd generation (+43 g, p<0.05) and the 2.5 generation mother (+54 g, p<0.001). By contrast, for the Raine Study, controlling for socioeconomic factors somewhat *decreased* children of immigrants' birth weight relative to children of natives, the few marginally significant differences became non-statistically significant. M3, which additionally controlled for mothers' smoking during pregnancy (black bars), showed a reduction in the magnitude of the coefficients for children of immigrants for both cohorts, which is still significant and positive for ELFE's 2.5 generation mother (+43 g, p<0.01).

The opposite pattern is observed in figure 2 M1, where children of immigrants exhibit lower birth weights compared with children of natives. This pattern characterises the BiB (Bradford, UK), the ABCD (Amsterdam, Netherlands) and the Generation R (Rotterdam, Netherlands) cohorts. Exceptions to this pattern are 2.5 generation children with an immigrant mother in the two Dutch cohorts, where no significant differences from natives' birth weight could be detected.

For the Dutch cohorts (ABCD and Generation R), the differences in birth weight for children with two immigrant parents and for those with one immigrant father are somewhat reduced after controlling for socioeconomic characteristics (M2), suggesting that the lower birth weight might be due to immigrants' disadvantaged socioeconomic position within the host country. By contrast socioeconomic factors do not seem to play an important role in explaining birth weight differences for the BiB (UK) cohort.[35]

As in the first group, mothers' smoking during pregnancy had a negative effect on children's birth weight. Despite mothers' lower prevalence of smoking during pregnancy, children of immigrants' still exhibit lower birth weight than children of natives after controlling for this variable. As shown in figure 2, for BiB birth weight gaps tend to increase in magnitude after controlling for smoking during pregnancy (M3, 2nd generation −119 g, p<0.001, 2.5 gen-mother −133 g, p<0.001 and 2.5 gen-father −148 g, p<0.001), indicating that if immigrant mothers smoked as much as native mothers, children of immigrants' birth weight would be even lower. For ABCD and Gen R, smoking has a smaller effect, but it slightly increased the gap for the second generation (−67 g, p<0.01 for ABCD, and −50 g, p<0.05 for Gen R). The mothers' height and pre-pregnancy weight are positively and similarly associated with the child's birth weight for all cohorts.

In sum, we find higher birth weight for children of immigrants in our southern European studies, such as in France and in the smaller cohorts representing Italy and Spain, as well as in Australia; and lower birth weight for children of immigrants in central and northern

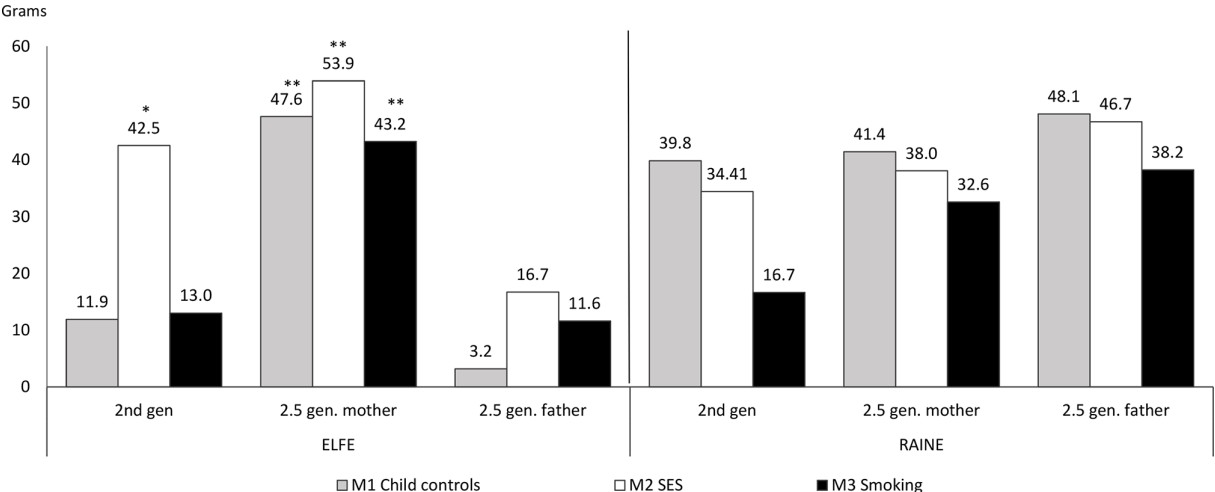

**Figure 1** Ordinary least square coefficients for cohorts in which children of immigrants exhibit higher birth weight (in grams). ELFE, Etude Longitudinale Française depuis l'Enfance; SES, socioeconomic status. *p <0.05, **p <0.01.

European countries, such as the UK and the Netherlands. These patterns could result from differences in the immigrants' regions of origin across host countries. We therefore analyse birth weight differences by maternal region of origin in the next section.

### Birth weight differences by maternal region of birth

Results from the models by mothers' region of origin are presented in table 3. First, we observe heterogeneity in birth weight by immigrant mothers' regions of birth. For ELFE (France), the pattern of birth weight 'advantage' and the changes across models are mostly driven by children of Middle East and North African immigrants, the largest immigrant group in France, as well as by children of East Asia and Pacific immigrants. However, children of sub-Saharan Africans, the second largest immigrant group in France do not appear to have statistically significantly different birth weights relative to children of

natives. Nonetheless, immigrant mothers exhibit lower smoking rates in most cohorts; thus, controlling for mother's smoking during pregnancy (table 3) explained part of the weight 'advantage' for two groups (East Asia and Pacific: +132 g, p<0.01; Middle East and North Africa: +61 g, p<0.01).

In the Raine Study, children of Western Europeans, the largest immigrant group, showed slightly higher birth weight than natives, but differences were not statistically significant. Children of North American mothers also reported heavier weights at birth (M1: +374 g, p<0.01; M2: +373 g, p<0.01, M3: +355 g, p<0.01), although these estimates are based on small sample sizes. Children with a South Asian and sub-Saharan's background tended to have lower birth weight than children of natives, but the difference was again not statistically significant.

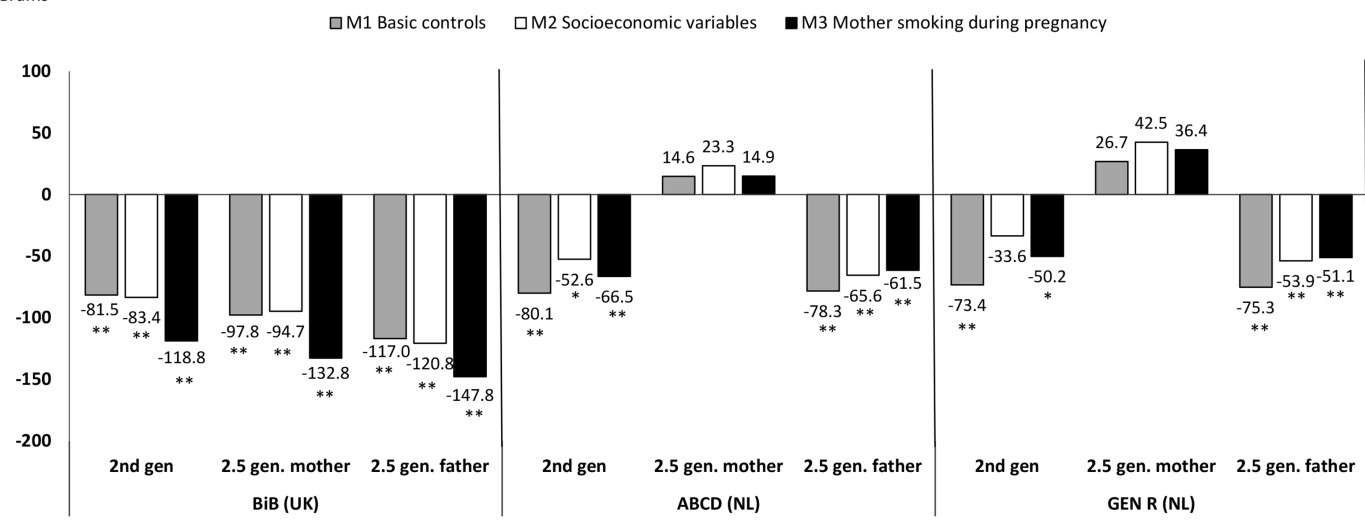

**Figure 2** Ordinary least square regression coefficients for cohorts in which children of immigrants exhibit lower birth weight (in grams). ABCD, Amsterdam Born Children and their Development; BiB, Born in Bradford; Gen R, Generation R; SES, socioeconomic status. *p <0.05, **p <0.01.

**Table 3** Ordinary least square regression coefficients of mother's region of origin on child's birth weight (large cohorts)

| DV: birth weight | ELFE (FR) | | | Raine Study (AU) | | | BiB (UK) | | |
|---|---|---|---|---|---|---|---|---|---|
| | M1 | M2 | M3 | M1 | M2 | M3 | M1 | M2 | M3 |
| | Child controls | SES | Smoked | Child controls | SES | Smoked | Child controls | SES | Smoked |
| Mother's region of origin | | | | | | | | | |
| Host country-born (ref.) | | | | | | | | | |
| Western EU/EEA | 24.7 | 21 | 20.8 | 37.2 | 30.3 | 25.3 | 45.1 | 42.9 | 30.3 |
| Eastern EU | 97.9 | 94.6 | 85.2 | 38.1 | 37.5 | 29.9 | 44.1 | 34.6 | 29.5 |
| Other Europe and Central Asia | 5.7 | 13.8 | 1 | 13.8 | 10.1 | −9 | −10.2 | 6.5 | −13.3 |
| East Asia and Pacific | 145.3** | 142.3** | 131.8** | −51.2 | −54.2 | −87.4 | 115.3 | 96.1 | 80.2 |
| South Asia | 211.9 | 211.1 | 244.7 | 40.2 | 40.8 | 36.5 | −74.1*** | −69.4*** | −94.4*** |
| Middle East and North Africa | 60.4** | 86.6*** | 61.2** | −35.8 | −46.9 | −67.9 | −35.3 | −45 | −60.5 |
| Sub-Saharan Africa | −38.5 | −9.8 | −40.1 | 165.7 | 165.7 | 126.6 | −53.8 | −58.1 | −82.1 |
| Latin America and Caribbean | −7.1 | 8.7 | −14.5 | | | | | | |
| North America | −67.6 | −76.6 | −56.7 | 374.4** | 373.1** | 354.9** | 108.7 | 86.4 | 70.2 |
| Child controls | | | | | | | | | |
| Female | −149.5*** | −149.2*** | −150.3*** | −128.2*** | −128.3*** | −122.4*** | −143.1*** | −141.4*** | −141.9*** |
| Plural birth† | −372.7*** | −374.2*** | −378.6*** | 0 | 0 | 0 | −319.7*** | −313.2*** | −318.7*** |
| Mother's parity (birth order) | 50.3*** | 55.3*** | 54.2*** | 64.7*** | 65.1*** | 64*** | 52.8*** | 59.9*** | 57*** |
| Gestational age | 23.7*** | 23.6*** | 23.5*** | 20.1*** | 20.2*** | 20.2*** | 25.4*** | 25.5*** | 25.3*** |
| Mother controls | | | | | | | | | |
| Height | 9.1*** | 8.5*** | 8.8*** | 10.6*** | 10.3*** | 10.6*** | 11.9*** | 11.7*** | 11.9*** |
| Pre-pregnancy weight | 4.7*** | 5*** | 4.9*** | 5.7*** | 5.8*** | 5.6*** | 5.6*** | 5.6*** | 5.6*** |
| SES | | | | | | | | | |
| Education‡ | | | | | | | | | |
| High (ref) | | | | | | | | | |
| Medium | | −11.8 | −3.2 | | 12.6 | 13.7 | | 0.1 | 2.2 |
| Low | | −34.4 | −15.1 | | −6.5 | 13.7 | | −17.4 | −13.6 |
| Household income quintiles§ | | | | | | | | | |
| 1st quintile (ref.) | | | | | | | | | |
| 2nd quintile | | 51.2*** | 43.8*** | | −4.8 | 7.6 | | 2.4 | −6.1 |
| 3rd quintile | | 49.6*** | 32.2* | | −35.3 | −15.1 | | −44.4 | −33.5 |
| 4th quintile | | 55.1*** | 34.4* | | −37.1 | 2.3 | | −43.7* | −35.7 |
| 5th quintile | | 58.4*** | 33.9* | | | | | −63.5** | −32.2 |

Continued

**Table 3** Continued

| DV: birth weight | ELFE (FR) | | | Raine Study (AU) | | | BiB (UK) | | |
|---|---|---|---|---|---|---|---|---|---|
| | M1 | M2 | M3 | M1 | M2 | M3 | M1 | M2 | M3 |
| | Child controls | SES | Smoked | Child controls | SES | Smoked | Child controls | SES | Smoked |
| Mother smoked (pregnancy) | | | −119.8*** | | | −157.5*** | | | −108.9*** |
| Constant | −5043*** | −4992*** | −4965*** | −4264*** | −4212*** | −4220*** | −6050*** | −5990*** | −5984*** |
| N | 12 419 | 12 419 | 12 419 | 2310 | 2310 | 2310 | 4141 | 4141 | 4141 |
| $R^2$ | 0.358 | 0.36 | 0.369 | 0.38 | 0.38 | 0.4 | 0.485 | 0.488 | 0.492 |

| DV: birth weight | ABCD (NL) | | | Gen R (NL) | | |
|---|---|---|---|---|---|---|
| | M1 | M2 | M3 | M1 | M2 | M3 |
| | Child controls | SES | Smoked | Child controls | SES | Smoked |
| Mother's region of origin | | | | | | |
| Host country-born (ref.) | | | | | | |
| Western EU/EEA | 7.8 | 10.5 | 0.8 | 32.3 | 34.3 | 40.5 |
| Eastern EU | −99.4 | −83.2 | −90.1 | 68.3 | 78.7 | 99.1 |
| Other Europe and Central Asia | 28.8 | 67.7 | 68.5 | 11.5 | 52.2 | 58.5 |
| East Asia and Pacific | 121.3* | 134.5* | 122.7* | 124** | 127.5** | 143.3** |
| South Asia | −174.8** | −144.3* | −170.9** | 11.4 | 9.4 | 35.8 |
| Middle East and North Africa | −6.2 | 25.7 | 1.7 | 20.5 | 38.8 | 63.3 |
| Sub-Saharan Africa | −54.6 | −19.9 | −41.1 | −115.8** | −86.9* | −68.5 |
| Latin America and Caribbean | −89.7** | −63* | −73.3* | −71.5** | −49.8* | −37.1 |
| North America | −148.5 | −143.1 | −151 | −62.5 | −74.8 | −64.6 |
| Child controls | | | | | | |
| Female | −121.7*** | −123.4*** | −124*** | −107.8*** | −109.1*** | −108.2*** |
| Plural birth† | 0 | 0 | 0 | −404.5*** | −409.4*** | −409*** |
| Mother's parity (birth order) | 80.3*** | 83.5*** | 82.3*** | 107.1*** | 108.7*** | 110.9*** |
| Gestational age | 25.7*** | 25.5*** | 25.5*** | 26.1*** | 25.9*** | 25.9*** |
| Mother controls | | | | | | |
| Height | 10.9*** | 9.7*** | 9.7*** | 11.4*** | 10.2*** | 10.2*** |
| Pre-pregnancy weight | 4.7*** | 5.3*** | 5.2*** | 4.6 | 4.9*** | 5*** |
| SES | | | | | | |
| Education‡ | | | | | | |
| High (ref) | | | | | | |
| Medium | | −66.3*** | −54.7*** | | −1.9 | −0.1 |

**Table 3** Continued

| DV: birth weight | ABCD (NL) | | | Gen R (NL) | | |
|---|---|---|---|---|---|---|
| | M1 | M2 | M3 | M1 | M2 | M3 |
| | Child controls | SES | Smoked | Child controls | SES | Smoked |
| Low | | −78.3*** | −56* | | −49.7 | −43.6 |
| Household income quintiles§ | | | | | | |
| 1st quintile (ref.) | | | | | | |
| 2nd quintile | | 0.5 | 3.9 | | 23.9 | 18 |
| 3rd quintile | | −10.9 | −0.5 | | 57.2** | 44.8* |
| 4th quintile | | −37.9 | −23.5 | | 93.6** | 73.4* |
| 5th quintile | | | | | 100.9** | 79.6* |
| Mother smoked (pregnancy) | | | −140.1*** | | | −77.3*** |
| Constant | −5827** | −5598*** | −5571*** | −6189*** | −6004*** | −5962*** |
| N | 4072 | 4072 | 4072 | 5123 | 5123 | 5123 |
| $R^2$ | 0.398 | 0.402 | 0.408 | 0.456 | 0.461 | 0.465 |

M1: basic child and mother's controls at birth. M2: M1 + socioeconomic status (SES) variables. M3: M2 + mother's smoking during pregnancy.

*p<0.05, **p<0.01, ***p<0.001.

†The ABCD and the Raine Study cohorts only included single births in their sample.

‡Level of education based on the highest ongoing or completed education when the child was 0 years old (between >−1 year and <1 year). If more than one education level is reported within the defined time frame, we used the highest recorded education level. Classification according to International Standard Classification of Education 97/2011 (ISCED-97/2011). High: short cycle tertiary, bachelor, masters, doctoral or equivalent (ISCED-2011: 5–8, ISCED-97: 5–6). Medium: upper secondary, post-secondary non-tertiary (ISCED-2011: 3–4, ISCED-97: 3–4). Low: no education; early childhood; pre-primary; primary; lower secondary or second stage of basic education (ISCED-2011: 0–2, ISCED-97: 0–2).

§European Union Statistics on Income and Living Conditions income quintiles for all cohorts except for Raine (AU) and ABCD (NL) that measured household income in quartiles.

ABCD, Amsterdam Born Children and their Development; AU, Australia; BiB, Born in Bradford; EEA, European Economic Area; ELFE, Etude Longitudinale Française depuis l'Enfance; EU, European Union; Gen R, Generation R.

For BiB, the observed pattern is driven by children of South Asian immigrants, the largest immigrant group in Bradford, who weighed less at birth than children of natives (M1: −74 g, p<0.001, M2: −69 g, p<0.001, M3: −94 g, p<0.001). By contrast, children of European and East Asia and Pacific as well as North American immigrants showed a higher, but non-significant, birth weight than children of natives. The weight disadvantage among children of South Asian immigrants was partly explained by socioeconomic status. Adjusting to mother's smoking during pregnancy exacerbated this disadvantage.

For ABCD, children of women from South Asia and Latin America and Caribbean exhibited significantly lower birth weight, even after controls. Children of East Asian mothers had a significantly higher birth weight after controlling for socioeconomic variables and mother's smoking.

For Generation R (Rotterdam, the Netherlands), both patterns also coexist: children of women born in sub-Saharan Africa and Latin America and Caribbean exhibited significantly lower birth weight, even after controls. Children of East Asian and of other European and Central Asian mothers had, on the contrary, significantly higher birth weights.

## DISCUSSION

Research in developed countries has identified an immigrant health advantage among adults,[5 7] yet, evidence on whether immigrants can transfer this health advantage to their children is mixed, and remains particularly scarce in the European context.[4 8 11 12] This study investigates whether children of first-generation immigrants exhibit a health advantage on weight at birth, comparing eight European countries and Australia.

Prior studies on the role of immigration on birth weight have focused on Latin-American immigrants in the USA, providing mixed evidence, for example, finding lighter birth weight among Mexican and Cuban babies and heavier birth weight among Puerto Rican babies relative to US-born white women.[8 9] Our research extends this area of research to European countries and Australia. The first aim of this study was to evaluate whether children of immigrants exhibit a higher or lower birth weight relative to children of natives, using recent harmonised panel data of rich host countries. Two broad patterns emerged: we found higher birth weight for immigrants' children in southern European and Australian cohort studies, and lower birth weights in northern European studies, relative to natives'. This finding questions, at least when considering birth health, whether the 'healthy migrant' narrative applies to all contexts, and calls for a more nuanced approach to study this phenomenon.

Our second aim was to explore in more detail birth weight variation by mother's country of origin, controlling for the mother's height and pre-pregnancy weight. Previous studies indicate that geographic origins could be associated with children of immigrants' birth weight.

For example, previous work has highlighted higher birth weights for infants born in Spain to African and Latin-American mothers,[12 36] while lower mean birth weights were observed for infants born in China to Asian Indian mothers.[37] One of the most interesting findings in our study is that children of South Asian mothers showed consistently lower birth weights, in both BiB-UK (where south Asian immigrants represent the largest immigrant group), ABCD-NL and to a lesser extent in the Raine Study-AU. South Asia bears half of global low birth weight burden; children of South Asian parents were consistently shown to have lower birth weight than natives in UK,[38] Australia and Netherlands. Another group that stands out includes East Asian and Pacific mothers' babies, who were significantly heavier at birth in three cohorts, France and the two Dutch cohorts. In France, the largest immigrant group, immigrants from the Middle East and North Africa, showed the strongest advantage in birth weight, mirroring similar results previously found in Belgium.[18 39] Birth weights of children with Latin-American and Caribbean mothers were significantly lower in the Dutch cohorts, in contrast to findings in previous work in Spain.[12 36]

Our third aim was to evaluate the impact of mother's smoking during her pregnancy. Our results seem to go in the same direction as previous studies,[20 22] finding that mother's smoking during pregnancy has a negative effect on birth weight, as tobacco being a vasoconstrictor reduces placental circulation. Immigrant mothers' lower prevalence of smoking during pregnancy in all study cohorts had therefore a protective effect, favouring their children's health at birth across countries.

The strength of our work lies in the use of the recent and harmonised cross-country EU Child Cohort Network data.[24] We are however not able to empirically measure other important contextual factors (potential confounders) that might drive some of these patterns and that differ across countries, such as access to health services, in both origin and host countries, access to employment, exposure to discrimination and to other health risk factors. Another methodological limitation is that the cohort samples are not all representative of the general population at the national level; some are local samples and some exclude specific groups (such as preterm babies in the Piccolipiù or ELFE cohorts). In addition, the most vulnerable and recently arrived immigrant groups, who are likely to have language difficulties and limited access to health institutions, may be under-represented in some cohort samples. If children in these groups are lighter at birth,[18 39] the current study may under-report low birth weight for children of immigrant, and thus overestimate their average birth weight. As a result, our findings can be better generalised to the largest nationally representative samples and longer established immigrant groups, and may be more limited for smaller and locally based samples, and newly arrived immigrant groups. Finally, the relationship between parental migration status and children's birth weight may change over time. This change is unlikely to meaningfully affect the

comparability of our findings between cohorts, as all our cohorts include births in the 2000s and early 2010s, expect for the Raine Study in which babies were born in 1989. Therefore, given the large difference in data collection period, the comparison between the European and Australian (Raine Study) results should be interpreted with caution.

## CONCLUSION

The patterns of birth weight of children of first-generation immigrants relative to natives differ across host countries. Some of this cross-country variation seems to be due to the diverse composition of immigrant communities across Europe and Australia. Further research should investigate whether these variations are also partly driven by the different social and health policies in host countries. Improving access to healthcare, especially during pregnancy, and more inclusive social policies are needed to reduce the inequalities in birth weight, especially for disadvantaged immigrant groups, while supporting positive parental health behaviours. Our results confirm the protective effects of not smoking during pregnancy for child's birth weight, highlighting the importance of maintaining immigrants' healthier practices.

**Author affiliations**
[1]French National Institute for Demographic Studies, INED, Paris, France
[2]Centre for Research on Social Inequalities (CRIS), Sciences Po, Paris, France
[3]UMR5193, LISST-CERS, Université Toulouse Jean Jaurès, Toulouse, France
[4]Department of Public Occupational Health, Amsterdam UMC, University of Amsterdam, Amsterdam Public Health Research Institute, Reproduction and Development Research Institute, Amsterdam, The Netherlands
[5]Nutrition and Health Innovation Research Institute, Edith Cowan University School of Medical and Health Sciences, Perth, Western Australia, Australia
[6]Telethon Kids Institute, School of Population and Global Health, University of Western Australia, Perth, Western Australia, Australia
[7]Sub Directorate for Public Health and Addictions of Gipuzkoa, Ministry of Health of the Basque Government, San Sebastián, Spain
[8]Group of Environmental Epidemiology and Child Development, Biodonostia Health Research Institute, San Sebastián, Spain
[9]Spanish Consortium for Research on Epidemiology and Public Health (CIBERESP), Instituto de Salud Carlos III, Madrid, Spain
[10]ISGlobal, Barcelona, Spain
[11]Agència de Salut Pública de Barcelona, ISGlobal, Barcelona, Spain
[12]Bradford Teaching Hospitals NHS Foundation Trust, Bradford Institute for Health Research, Bradford, UK
[13]Department of Epidemiology, GECKO Drenthe Cohort, University Medical Center Groningen, University of Groningen, Groningen, The Netherlands
[14]Dipartimento di Scienze Mediche, Universita degli Studi di Torino, Torino, Italy
[15]Department of Medical Sciences, University of Turin, Torino, Italy
[16]The Generation R Study Group, University Medical Center, Erasmus Medical Center, Rotterdam, The Netherlands
[17]University Medical Center, Erasmus Medical Center Department of General Pediatrics, Rotterdam, The Netherlands
[18]Unit of Epidemiology, Meyer Children's University Hospital, Florence, Italy
[19]Predepartamental Unit of Medicine, Universitat Jaume I, Castello de la Plana, Comunitat Valenciana, Spain
[20]CIBERESP, Madrid, Spain
[21]Joint Research Unit in Epidemiology, Environment and Health, FISABIO, Valencia, Spain
[22]Clinical Epidemiology and Public Health Research Unit, Istituto di Ricovero e Cura a Carattere Scientifico materno infantile Burlo Garofolo, Trieste, Italy
[23]Inserm and INED Joint Research Group, Paris, France
[24]Université Paris Cité, Inserm, Inrae, Cress, Paris, France

**Acknowledgements** The ELFE survey is a joint project between the French Institute for Demographic Studies (INED) and the National Institute of Health and Medical Research (INSERM), in partnership with the French blood transfusion service (Etablissement français du sang, EFS), Santé publique France, the National Institute for Statistics and Economic Studies (INSEE), the Direction générale de la santé (DGS, part of the Ministry of Health and Social Affairs), the Direction générale de la prévention des risques (DGPR, Ministry for the Environment), the Direction de la recherche, des études, de l'évaluation et des statistiques (DREES, Ministry of Health and Social Affairs), the Département des études, de la prospective et des statistiques (DEPS, Ministry of Culture), and the Caisse nationale des allocations familiales (CNAF), with the support of the Ministry of Higher Education and Research and the Institut national de la jeunesse et de l'éducation populaire (INJEP). The ELFE cohort also acknowledges the support from F Zariouh, M A Bellance, M Cornet and B de Lauzon-Guillain in harmonising the ELFE cohort data for the LifeCycle Project. The authors would like to acknowledge the Raine study participants and their families for their ongoing participation in the study and the Raine study team for study coordination and data collection. We also thank the NHMRC for their long-term contribution to funding the study over the last 30 years. The core management of the Raine study is funded by The University of Western Australia, Curtin University, Telethon Kids Institute, Women and Infants Research Foundation, Edith Cowan University, Murdoch University, The University of Notre Dame Australia and the Raine Medical Research Foundation. Born in Bradford is only possible because of the enthusiasm and commitment of the Children and Parents in BiB. We are grateful to all the participants, practitioners and researchers who have made Born in Bradford happen. The Generation R study is conducted by the Erasmus Medical Center in close collaboration with the School of Law and Faculty of Social Sciences of the Erasmus University Rotterdam, the Municipal Health Service Rotterdam area, Rotterdam, the Rotterdam Homecare Foundation, Rotterdam and the Stichting Trombosedienst & Artsenlaboratorium Rijnmond (STAR-MDC), Rotterdam. We gratefully acknowledge the contribution of children and parents, general practitioners, hospitals, midwives and pharmacies in Rotterdam. We also acknowledge The University of Western Australia, Curtin University, Women and Infants Research Foundation, Telethon Kids Institute, Edith Cowan University, Murdoch University, The University of Notre Dame Australia and The Raine Medical Research Foundation for providing funding for the core management of the Raine study. INMA-Gipuzkoa is grateful to the participants of the study for their kind collaboration. The recruitment of participants and the development of the study was funded by grants from Instituto de Salud Carlos III (PI06/0867 and FIS-PI09/00090 incl. FEDER funds), Department of Health of the Basque Government (2005111093), Provincial Government of Gipuzkoa (DFG06/002), and annual agreements with the municipalities of the study area (Zumarraga, Urretxu, Legazpi, Azkoitia y Azpeitia y Beasain). We are grateful to all the participating children, parents, practitioners and researchers who took part in the INMA Sabadell study. We acknowledge support from the Spanish Ministry of Science and Innovation through the 'Centro de Excelencia Severo Ochoa 2019-2023' Programme (CEX2018-000806-S), and support from the Generalitat de Catalunya through the CERCA Programme. The authors acknowledge the Piccolipiù Working Group and the families involved in the study. The Childhood Obesity Project (CHOP) thanks the participating families and all project partners for their enthusiastic support of the project. Furthermore, thanks to the European Childhood Obesity Trial Study Group, who designed and conducted the study, entered the data and participated in the data analysis. The authors are grateful to all the participants of the NINFEA birth cohort. We are grateful to the families who took part in the GECKO Drenthe study, the midwives, gynaecologists, nurses and GPs for their help for recruitment and measurement of participants, and the whole team from the GECKO Drenthe study.

**Contributors** LP and MI conceived the idea for the study. VJ supervised the collaboration project for all LifeCycle cohorts. SF conducted the analyses for the EFLE cohort. SF, MI and LP wrote the statistical analysis plan. LP, MI, SF and M-AC contributed to the interpretation of results for ELFE. JW monitored data collection for the BiB study. TCY and JW analysed the data and contributed to the interpretation of the findings for the BiB study. MC and EC conducted the analysis and interpreted the results for GECKO. MW, EV, VJ and MNK and conducted the analysis for the Generation R study. TGMV and MWH-vG conducted the analysis and interpreted the results for the ABCD cohort. R-CH and JC conducted the analyses and interpreted the results for the Raine study. SF-B, LSMR, MS-P and MV analysed data for the INMA cohort. EI, CM and LR participated in the analysis and interpretation of data for NINFEA, whereas ES, MR and AB did so for the Piccolipiù cohort. SF and SP-J coordinated the analyses among all cohorts. SF drafted the manuscript with the feedback of LP and MI. All authors revised

the manuscript. All authors approved the final version of the paper. SF is the guarantor.

**Funding** The LifeCycle Project received funding from the European Union's Horizon 2020 research and innovation programme (grant agreement no. 733206 LifeCycle). The ELFE study received a government grant managed by the National Research Agency under the 'Investissements d'avenir' programme (ANR-11-EQPX-0038). Core funding for Born in Bradford (BiB) has been funded by the Wellcome Trust (WT101597MA), a joint grant from the UK Medical Research Council (MRC) and UK Economic and Social Science Research Council (ESRC) (MR/N024397/1); the British Heart Foundation (CS/16/4/32482) and the National Institute for Health Research ARC Yorkshire and Humber (NIHR200166). The views expressed in this publication are those of the author(s) and not necessarily those of the National Institute for Health Research or the Department of Health and Social Care or Wellcome Trust. The GECKO Drenthe birth cohort was funded by an unrestricted grant of Hutchison Whampoa Ld, Hong Kong and supported by the University of Groningen, Well Baby Clinic Foundation Icare, Noordlease, Paediatric Association Of The Netherlands, Youth Health Care Drenthe and European Union's Horizon 2020 research and innovation programme (grant agreement no. 733206 LifeCycle). The Raine study has been funded by programme and project grants from the Australian National Health and Medical Research Council, the Commonwealth Scientific and Industrial Research Organisation, Healthway and the Lions Eye Institute in Western Australia. The Raine study Gen2-17 years of follow-up was funded by the NHMRC Programme Grant (Stanley *et al*, ID 353514). The Raine study participation in LifeCycle was funded by a grant from the National Health and Medical Research Council, Australia (GNT1142858). The University of Western Australia (UWA), Curtin University, the Raine Medical Research Foundation, the Telethon Kids Institute, the Women's and Infant's Research Foundation (KEMH), Murdoch University, The University of Notre Dame Australia and Edith Cowan University provide funding for the Core Management of the Raine study. The general design of the Generation R study is made possible by financial support from the Erasmus MC, University Medical Center, Rotterdam, Erasmus University Rotterdam, Netherlands Organisation for Health Research and Development (ZonMw), Netherlands Organisation for Scientific Research (NWO), Ministry of Health, Welfare and Sport and Ministry of Youth and Families. This project received funding from the European Union's Horizon 2020 research and innovation programme (LifeCycle, grant agreement no. 733206, 2016; EUCAN-Connect grant agreement no. 824989; ATHLETE, grant agreement no. 874583). VJ received funding from a Consolidator Grant from the European Research Council (ERC-2014-CoG-648916). INMA-Sabadell data collections were supported by grants from the Instituto de Salud Carlos III (Red INMA G03/176), CIBERESP, and the Generalitat de Catalunya-CIRIT (1999SGR 00241). The NINFEA cohort was partially funded by the Compagnia San Paolo and by the Piedmont Region. The Piccolipiù project was financially supported by the Italian National Center for Disease Prevention and Control (CCM grants years 2010 and 2014) and by the Italian Ministry of Health (art 12 and 12 bis D.lgs 502/92). Childhood Obesity Project (CHOP): the study reported has been carried out with partial financial support from the Commission of the European Community; specific research, technological development and demonstration programme 'Quality of Life and Management of Living Resources', within the European Union's Seventh Framework Programme (FP7/2007-2013); project EarlyNutrition under grant agreement no. 289346; the European Union H2020 project LifeCycle under grant no. 733206; the European Research Council Advanced Grant META-GROWTH (ERC-2012-AdG: no. 322605) and partial financial support from the Polish Ministry of Science and Higher Education (2571/7.PR/2012/2).

**Competing interests** None declared.

**Patient and public involvement** Patients and/or the public were not involved in the design, or conduct, or reporting, or dissemination plans of this research.

**Patient consent for publication** Not applicable.

**Ethics approval** Not applicable.

**Provenance and peer review** Not commissioned; externally peer reviewed.

**Data availability statement** Data may be obtained from a third party and are not publicly available.

**ORCID iDs**
Sandra Florian http://orcid.org/0000-0001-8553-6485
Margreet W Harskamp-van Ginkel http://orcid.org/0000-0001-8792-6663
Jennie Carson http://orcid.org/0000-0001-5294-7536
Tiffany C Yang http://orcid.org/0000-0003-4549-7850
John Wright http://orcid.org/0000-0001-9572-7293
Elena Isaevska http://orcid.org/0000-0002-1846-7990

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
