## [Reviewer comments · BMJ Open]

ARTICLE DETAILS

TITLE (PROVISIONAL)	Differences in birthweight between immigrants' and natives' children in Europe and Australia: A LifeCycle comparative observational cohort study
AUTHORS	Florian, Sandra; Ichou, Mathieu; Panico, Lidia; Pinel-Jacquemin, Stéphanie; Vrijkotte, Tanja G. M.; Harskamp-van Ginkel, Margreet; Huang, Rae-Chi; Carson, Jennie; Rodriguez, Loreto; Subiza-Pérez, Mikel; Vrijheid, Martine; Fernandez, Sílvia; Yang, Tiffany; Wright, John; Corpeleijn, Eva; Cardol, Marloes; Isaevska, Elena; Moccia, Chiara; Kooijman, Marjolein; Voerman, Ellis; Jaddoe, Vincent; Welten, Marieke; Spada, Elena; REBAGLIATO, MARISA; Beneito, Andrea; Ronfani, Luca; Charles, Marie-Aline

VERSION 1 – REVIEW

REVIEWER	C Bréhin Centre Hospitalier Universitaire de Toulouse
REVIEW RETURNED	17-Feb-2022

GENERAL COMMENTS	This multicenter study asks the question of the “immigrant health paradox” on child health by taking child birth weight as the primary outcome. This study is very interesting in its methodology because it includes several cohort studies in different countries with different immigration origins; Here are some remarks: Introduction: Page 3 line 15 to 32: The state of the art on the “immigrant health paradox” on adults and children could be better developed by adding these references: Song IG, Kim MS, Shin SH, Kim EK, Kim HS, Choi S, Kwon S, Park SM. Birth outcomes of immigrant women married to native men in the Republic of Korea: a population register-based study. BMJ Open. 2017 Sep 24;7(9):e017720. doi: 10.1136/bmjopen-2017-017720. PMID: 28947460; PMCID: PMC5623517. Guillot M, Khat M, Elo I, Solignac M, Wallace M. Understanding age variations in the migrant mortality advantage: An international comparative perspective. PLoS One. 2018 Jun 29;13(6):e0199669. doi: 10.1371/journal.pone.0199669. PMID: 29958274; PMCID: PMC6025872. Ramraj C, Pulver A, Siddiqi A. Intergenerational transmission of the healthy immigrant effect (HIE) through birth weight: A systematic review and meta-analysis. Soc Sci Med. 2015 Dec;146:29-40. doi: 10.1016/j.socscimed.2015.10.023. Epub 2015 Oct 22. PMID: 26492459. Kelagher M, Jessop DJ. Differences in low-birthweight among documented and undocumented foreign-born and US-born Latinas.
---

	Soc Sci Med. 2002 Dec;55(12):2171-5. doi: 10.1016/s0277-9536(01)00360-4. PMID: 12409130. Restrepo-Mesa SL, Estrada-Restrepo A, González-Zapata LI, Agudelo-Suárez AA. Newborn birth weights and related factors of native and immigrant residents of Spain. J Immigr Minor Health. 2015 Apr;17(2):339-48. doi: 10.1007/s10903-014-0089-5. PMID: 25150559. Racape J, Schoenborn C, Sow M, Alexander S, De Spiegelaere M. Are all immigrant mothers really at risk of low birth weight and perinatal mortality? The crucial role of socio-economic status. BMC Pregnancy Childbirth. 2016 Apr 8;16:75. doi: 10.1186/s12884-016-0860-9. PMID: 27059448; PMCID: PMC4826554. Reference 4 which is a study in children does not fit with the sentence "Research on adults has identified an immigrant health advantage, known as the "immigrant health paradox". Methods: Page 5 line 33 "Table 1 presents the sample distribution of parental migration status and maternal region of birth for these five cohorts" => to put in the results section Results: Table 2: Difficult reading because table in 2 parts, it would be necessary to give the first column with the studied determinants (Page 9) for the 2nd part of the table (page 10). Page 11: line 26-35: Specify that it is Model 1 Add a sentence about the association with the mother's height, weight before pregnancy and the child's birth weight (Table 2 and 3). Table 3: same remark as for table 2 Missing data are not clearly shown in the results or in the tables Discussion: - Add and expand the discussion with the references proposed in the introduction - Discuss the possible bias related to the different inclusion periods in the different cohorts Conclusion: Page 21 line 6: "it is likely that these differences are also partly driven by variations in social and health policies in the host countries": This is not a conclusion from the results of your study but a research hypothesis that is not explored in your study.
--	--

REVIEWER	Uğurcan Sayılı Republic of Turkey Ministry of Health, Karakopru District Health Directorate
REVIEW RETURNED	04-Mar-2022

GENERAL COMMENTS	The article touches on a very important issue. It is well written. But I have a major concern about statistics. Small differences can be statistically significant when studied with large sample sizes. For example, in the Netherlands -83 gr. UK: -82 g. Is this really a clinically significant difference? First, I recommend that authors present a table and descriptive data. E.g: Infant birth weights in grams for natives and immigrants of each country. not only difference. The immigrant health effect is not visible globally. When I read your article and saw the statistics, I think the same thing as you. But from a different perspective. In the Netherlands -83 gr. UK: -82 g. These differences are not significant clinical differences; depends on the
---

	sample size. I recommend that a statistics editor review the article. If the statistics are not appropriate, all discussion section can be change. Therefore, I don't make any further evaluations. After the statistics editor's comment, i can make further evaluation.
--	---

REVIEWER	Joseph Nolan Northern Kentucky University, Mathematics & Statistics
REVIEW RETURNED	08-Nov-2022

GENERAL COMMENTS	This is primarily a statistical review. I'm as yet unconvinced as to the choice of model here (see point #2 below), though once the authors settle on that choice and provide clear justification, I think most other revisions should be minor. 1. Page 7, line 46: What is meant by a "complete-case analysis approach"? If this means that some datalines were not used because they were incomplete, you should provide specifics regarding the missing data and any potential impact on analysis. 2. Page 7, lines 46-56: It is curious, based on discussion of both low and high birthweights having higher risk per previous studies, (1) why is it appropriate to exclude babies having birthweight >4500 grams? If there is a nonlinear relationship, it seems like one should model that, rather than excluding and potentially adding experimenter bias. (2) For the logistic model, why not model it as "higher risk" birthweight (<2500 or >4500) vs. "normal" birthweight (or have three separate categories). It seems that including >4500 in with 2500-4500 for that model also might result in bias. Generally speaking, the model should be selected in advance of looking at the data, and should be based on scientific knowledge at that point. Based on what is indicated in the introduction, something along the lines of "high risk" vs. "normal", using logistic regression, seems most justifiable to me. Further, the model chosen for presentation should generally be chosen prior to conducting the study - running three models and choosing the one you like best is effectively data snooping and that "strategy" should generally be avoided. Table 2: The format is not acceptable. The table is obviously substantially large that it must cross multiple pages. Each page / sub-table should have both ROW and column headings. You will save space if you use superscripts rather than a separate column for p-values. Do NOT include $p < 0.10$. You are doing so many tests here, that ultimately some adjustment for multiplicity should likely be considered (discuss, or more likely include as a limitation if this was not done). You'd expect one in ten p-values to be < 0.10 just by random chance. Table 3 - all comments for Table 2 apply here as well.
--

VERSION 1 – AUTHOR RESPONSE

Reviewer: 1
Dr. C Bréhin, Centre Hospitalier Universitaire de Toulouse

Page 3 line 15 to 32: The state of the art on the “immigrant health paradox” on adults and children could be better developed by adding these references:

=> We appreciate the reference suggestions. Four of the six references suggested by Reviewer 1 have been added to the paper:

- Guillot M, Khlal M, Elo I, Solignac M, Wallace M. Understanding age variations in the migrant mortality advantage: An international comparative perspective. *PLoS One*. 2018 Jun 29;13(6):e0199669. doi: 10.1371/journal.pone.0199669. PMID: 29958274; PMCID: PMC6025872.
- Ramraj C, Pulver A, Siddiqi A. Intergenerational transmission of the healthy immigrant effect (HIE) through birth weight: A systematic review and meta-analysis. *Soc Sci Med*. 2015 Dec;146:29-40. doi: 10.1016/j.socscimed.2015.10.023. Epub 2015 Oct 22. PMID: 26492459.
- Restrepo-Mesa SL, Estrada-Restrepo A, González-Zapata LI, Agudelo-Suárez AA. Newborn birth weights and related factors of native and immigrant residents of Spain. *J Immigr Minor Health*. 2015 Apr;17(2):339-48. doi: 10.1007/s10903-014-0089-5. PMID: 25150559.
- Racape J, Schoenborn C, Sow M, Alexander S, De Spiegelaere M. Are all immigrant mothers really at risk of low birth weight and perinatal mortality? The crucial role of socio-economic status. *BMC Pregnancy Childbirth*. 2016 Apr 8;16:75. doi: 10.1186/s12884-016-0860-9. PMID: 27059448; PMCID: PMC4826554.

=> We were not able to include two of the suggestions; we give an explanation for this decision below each of the references:

- Song IG, Kim MS, Shin SH, Kim EK, Kim HS, Choi S, Kwon S, Park SM. Birth outcomes of immigrant women married to native men in the Republic of Korea: a population register-based study. *BMJ Open*. 2017 Sep 24;7(9):e017720. doi: 10.1136/bmjopen-2017-017720. PMID: 28947460; PMCID: PMC5623517.

=> This article focuses on a specific population (migrant women married to Korean men vs. migrant women married to non-Korean men, living in Korea), and includes an analysis of women from countries such as China, Philippines, Vietnam. Unfortunately, our samples for those origin countries are very limited.

- Kelaher M, Jessop DJ. Differences in low-birthweight among documented and undocumented foreign-born and US-born Latinas. *Soc Sci Med*. 2002 Dec;55(12):2171-5. doi: 10.1016/s0277-9536(01)00360-4. PMID: 12409130.

=> This paper focuses on documented vs. undocumented immigrants. We found that it was only tangential to our paper. The focus on Latina mothers is also not very relevant for the present paper.

Reference 4 which is a study in children does not fit with the sentence "Research on adults has identified an immigrant health advantage, known as the "immigrant health paradox".

=> We thank the reviewer for picking up this mistake. The reference has been moved to its correct location, in the previous paragraph of the introduction.

Page 5 line 33 “Table 1 presents the sample distribution of parental migration status and maternal region of birth for these five cohorts”, to put in the results section

=> This sentence has been moved to the Results section (from p.6 to p.10)

Results:

Table 2: Difficult reading because table in 2 parts, it would be necessary to give the first column with the studied determinants (Page 9) for the 2nd part of the table (page 10).

Table 3: same remark as for table 2

=> Tables 2 and 3 have been reformatted.

Page 11: line 26-35: Specify that it is Model 1

=> We have specified that those results relate to Model 1 (p.9)

Add a sentence about the association with the mother's height, weight before pregnancy and the child's birth weight (Table 2 and 3).

=> We have added a sentence on the positive association between the mother's height and pre-pregnancy weight and the child's birthweight on p. 10. We only mention it once (Table 2) to avoid repetition, as the associations are similar in Table 3.

Missing data are not clearly shown in the results or in the tables

=> A distribution of missing data has been added for all cohorts (Table 1 & Appendix 1).

Discussion:

- Add and expand the discussion with the references proposed in the introduction

=> We have expanded the discussion and added the proposed references.

- Discuss the possible bias related to the different inclusion periods in the different cohorts

=> This point is now addressed at the end of the discussion, noting that the comparison between the Australian (RAINE) results – in which babies were born in 1989 – and all the other cohorts – in which babies were born in the 2000s and early 2010s – should be interpreted with caution (p.14).

Conclusion:

Page 21 line 6: "it is likely that these differences are also partly driven by variations in social and health policies in the host countries": This is not a conclusion from the results of your study but a research hypothesis that is not explored in your study.

=> We appreciate this observation. We have reworded this sentence, which is now presented as an opening for future research.

Reviewer: 2

Dr. Uğurcan Sayılı, Republic of Turkey Ministry of Health

Small differences can be statistically significant when studied with large sample sizes.

For example, in the Netherlands -83 gr. UK: -82 g. Is this really a clinically significant difference?

These differences are not significant clinical differences; depends on the sample size.

=> While these differences might not be clinically substantive at an individual level, at a population/group level, statistically significant differences of 50 to 100 grams are indeed very large and speak to large differences in birthweight across these groups.

First, I recommend that authors present a table and descriptive data.

E.g: Infant birth weights in grams for natives and immigrants of each country. not only difference.

=> As recommended by Reviewer 2, we have included the birthweight for natives and immigrants for each cohort in Table 1 and Appendix 1.

The immigrant health effect is not visible globally. When I read your article and saw the statistics, I think the same thing as you. But from a different perspective. In the Netherlands -83 gr. UK: -82 g.

=> We agree with the reviewer. Indeed, as we mention in the discussion, much of this literature has been informed by a relatively small number of countries (both host and origin). We therefore believe that studies that increase the diversity of the observed populations are crucial to a more nuanced approach. We now make this point more explicitly in the discussion.

Reviewer: 3

Dr. J Nolan, Northern Kentucky University

This is primarily a statistical review. I'm as yet unconvinced as to the choice of model here (see point #2 below), though once the authors settle on that choice and provide clear justification, I think most other revisions should be minor.

1. Page 7, line 46: What is meant by a "complete-case analysis approach"? If this means that some datalines were not used because they were incomplete, you should provide specifics regarding the missing data and any potential impact on analysis.

=> This is an important observation. As Reviewer 3 correctly infers, by a "complete-case analysis approach" we mean that cases with missing values were excluded from the analyses. Following Reviewer 3's suggestion, we are now including the distribution of cases with missing data for each cohort in Table 1 (and Appendix 1). We agree that a complete case analysis is likely to exclude more disadvantaged, mobile households that are less likely to fully respond to questionnaires, such as disadvantaged immigrant groups, and therefore, we are likely underestimating the gap in countries where migrants appear to have worse health and overestimating the gap in countries where migrants report better outcomes. This is now noted in the Discussion section.

2. Page 7, lines 46-56: It is curious, based on discussion of both low and high birthweights having higher risk per previous studies, (1) why is it appropriate to exclude babies having birthweight >4500 grams? If there is a nonlinear relationship, it seems like one should model that, rather than excluding and potentially adding experimenter bias. (2) For the logistic model, why not model it as "higher risk" birthweight (<2500 or >4500) vs. "normal" birthweight (or have three separate categories). It seems that including >4500 in with 2500-4500 for that model also might result in bias.

Generally speaking, the model should be selected in advance of looking at the data, and should be based on scientific knowledge at that point. Based on what is indicated in the introduction, something along the lines of "high risk" vs. "normal", using logistic regression, seems most justifiable to me. Further, the model chosen for presentation should generally be chosen prior to conducting the study -

running three models and choosing the one you like best is effectively data snooping and that "strategy" should generally be avoided.

=> We appreciate Reviewer 3's remark and we would like to take the opportunity to clarify this issue. (1) We tried modeling the nonlinearity using a multinomial model with three categories (very low, normal and very high birthweight), yet for the largest cohorts the main findings did not change. Some other cohorts did not have enough cases with high birthweight, and thus lacked statistical power to run multinomial models. Thus, we ran the analyses with and without very high weight babies (see Appendix 3). The results were very similar including or excluding these babies, indicating that the relationship between birthweight and migrant status is pretty much linear, in any case, the potential nonlinearity is not large enough to alter the results presented in this study. Thus, the more parsimonious model that treats birthweight as linear was preferred. (2) Reviewer 3 is right, both very low and very high birthweights represent health risks for children. Yet, because the determinants of these two abnormal - very low and very high - birthweights are considerably different, they cannot be combined into the same category. For this reason, we also considered a logistic model predicting low birthweight, the results are presented in Appendix 4.

Table 2: The format is not acceptable. The table is obviously substantially large that it must cross multiple pages. Each page / sub-table should have both ROW and column headings. You will save space if you use superscripts rather than a separate column for p-values. Do NOT include $p < 0.10$. Table 3 - all comments for Table 2 apply here as well.

=> Tables 2 and 3 have been reformatted. As per Reviewer 3's request, we are no longer including $p < 0.10$.

VERSION 2 – REVIEW

REVIEWER	Joseph Nolan Northern Kentucky University, Mathematics & Statistics
REVIEW RETURNED	03-Feb-2023
GENERAL COMMENTS	The manuscript has been amended to address all of my primary concerns from the previous review, and from a statistical perspective seems sufficient for publication with one editorial change. In Table 1, a few numbers have been reported to extra decimal places in various places (e.g. 3321.036, 80.96). Please consistently report each column to 0 or 1 decimal place as you deem appropriate. Please review other tables for this potential inconsistency as well.